# Arctic Climate: Changes in Sea Ice Extent Outweigh Changes in Snow Cover

Aaron Letterly[1], Jeffrey Key[2], Yinghui Liu[1]

[1]Cooperative Institute for Meteorological Satellite Studies, University of Wisconsin, Madison, WI
[2] National Oceanic and Atmospheric Administration, Madison, WI

*Correspondence to*: Aaron Letterly (letterly@wisc.edu)

**Abstract.** Recent declines in Arctic sea ice and snow extent have led to an increase in the absorption of solar energy at the surface, resulting in additional surface heating and a further decline in snow and ice. Using 34 years of satellite data, 1982 - 2015, we found that the positive trend of solar absorption over the Arctic Ocean is more than double that over Arctic land, and the magnitude of the ice-albedo feedback is four times that of the snow-albedo feedback in summer. The timing of the high to low albedo transition has shifted closer to the greater insolation of the summer solstice over ocean, but further away from the summer solstice over land. Therefore, decreasing sea ice cover, not changes in terrestrial snow cover, has been the dominant radiative feedback mechanism over the last few decades.

## 1 Introduction

Over the last few decades satellites have observed an unprecedented reduction in Arctic sea ice extent (Pistone et al., 2014; Parkinson et al., 1999; Stroeve et al., 2012). Sea ice extent has decreased dramatically, with the ten lowest minimum Arctic sea ice extents after 2007. The Arctic-wide melt season has become longer from 1979 to 2013 with a rate of 5 days per decade (Stroeve et al., 2014). September sea ice extent decreased by 45% from 1979 to 2016, and if current trends continue, some Arctic shelf seas are forecasted to be ice-free during summer in the 2020s (Onarheim et al., 2018). Over Northern Hemispheric land, snow cover extent has been decreasing in all seasons (Hori et al., 2017). Shrinking sea ice cover and terrestrial snow cover decrease the reflectivity (albedo) of the surface, resulting in more absorption of solar (shortwave) radiation, more surface heating, and further reductions in snow and ice. These processes are known as the sea ice-albedo feedback over ocean and the snow-albedo feedback over land. Here we examine how changes in surface albedo over the ocean and land areas of the Arctic have affected shortwave absorption differently, and how this interplay between albedo and shortwave absorption may change in the future. Results are presented for the majority of the satellite record, from 1982 to 2015, and for the pan-Arctic from 60°N latitude to the pole.

Between 1979 and 2011, the Arctic top-of-atmosphere (planetary) albedo decreased from 0.52 to 0.48, and subsequent years with record or near-record low sea ice extent have further increased the amount of heat absorbed in the Arctic (Pistone et al., 2014). As the multi-year ice concentration decreases and is replaced by open water in the summer and thin, first-year ice in the winter, the darker surfaces reflect less sunlight and absorb more energy. The total absorbed solar radiation for the Arctic Ocean has therefore increased. Pinker et al. (2014) and Kashiwase et al. (2017) examined shortwave absorption in the upper Arctic ocean, with the latter finding that increases in open water may have led to a 50% increase in absorption since 1979.

The recent decreases in Arctic albedo are not entirely due to reduced sea ice cover, but also due to changes in the terrestrial snow cover (Robinson & Frei, 2000). Snow extent has decreased over Eurasia and North America since the late 1980s (Robinson & Frei, 2000; Kato et al., 2006) and is expected to continue decreasing by 3.7% (±1.1%) per decade during the spring over the 21st century (Thackeray et al., 2016). Hemispheric snow extent may strongly influence early spring temperatures through a strong positive feedback between spring snow cover and the radiative balance over mid- and high-latitude land in the Northern Hemisphere (Groisman et al., 1994), in which retreating snow cover has led to a lower polar albedo and increased radiative absorption in April and May over the satellite record (Robinson & Frei, 2000; Robinson et al., 2005). Since 2007, the decrease in Northern Hemisphere snow cover has accelerated during the late spring and summer due to warmer spring air temperatures augmenting surface net radiation (Hernández-Henríquez et al., 2015).

Though the radiative effects of reduced snow and ice cover are straightforward, changing surface types in the Arctic may initiate albedo interactions that are complex. More open water in the Arctic Ocean has also led to an increase in cloud cover (Liu et al., 2012), which could offset the decreases in summer albedo caused by melting ice (Kato et al., 2006) and the replacement of multiyear ice with thinner, first-year ice (Nghiem et al., 2007). In winter, when clouds inhibit radiative cooling of ice and open water, large anomalies in cloud cover may enhance or deter refreezing. This preconditioning of sea ice in the winter can influence the initial ice conditions for the spring melt and affect sea ice concentration (and therefore the Arctic albedo) through the melting season and into the fall of the following year (Letterly et al., 2016; Liu & Key, 2014).

The radiative feedbacks of changing snow cover and sea ice in the Northern Hemisphere have been studied (Perovich & Light, 2015; Fernandes et al., 2009; Flanner et al., 2011; Perovich et al., 2007). Perovich et al. (2007) analyzed the changes in solar energy during the melting period in the Arctic, but only over the period 1998-2004. Flanner et al. used top-of-atmosphere (TOA) fluxes to determine that the total impact of the cryosphere on radiative forcing between 1979 and 2008 was -4.6 to -2.2 Wm$^{-2}$. Their results included changes in snow and ice over the entire Northern Hemisphere, but applied a fixed annual albedo cycle over sea ice.

With satellite-derived surface radiative flux data now available from the early 1980s, it is now possible to study the relative effects of changing snow cover and sea ice on the Arctic surface energy budget. Does the increasingly early arrival of snow

melt in the spring reduce the Arctic surface albedo more than the decrease in sea ice during the summer? Have the climatological changes associated with a warming Arctic affected the absorption of solar radiation more over land or over sea? Will trends in Arctic land and ocean surface albedo result in similar trends in solar radiation absorption in the near future? In this study, we use satellite-derived surface radiative fluxes from 1982 to 2015 to examine the inter-annual changes in surface albedo and the absorption of solar energy caused by the timing of the melt onset, and to estimate the major albedo feedbacks from the ocean and land. This study focuses on the effects of snow and ice cover changes on the surface shortwave radiation budget of the Arctic - defined as the area poleward of 60°N - not the remote effects of mid-latitudes on the Arctic.

## 2 Arctic Shortwave Absorption Trends over Snow and Sea Ice

The primary dataset for this study is the Advanced Very High Resolution Radiometer (AVHRR) Polar Pathfinder Extended (APP-x) (Key et al., 2016). APP-x consists of twice-daily, 25 km composites at two local solar times in the Arctic (04:00 and 14:00) and Antarctic (02:00 and 14:00) starting in 1982. Data from 1982 through 2015 at 14:00 local solar time (high sun) are employed. APP-x includes surface temperature, surface broadband albedo, sea ice thickness, cloud properties (coverage, optical depth, effective particle size, thermodynamic phase, and top pressure), and radiative fluxes at the surface and TOA. In APP-x, the retrieval of surface albedo involves four steps. First, the reflectances of the two shortwave channels are converted to a broadband reflectance. Then, the TOA broadband reflectance is corrected for anisotropy and atmospheric attenuation, and converted from TOA broadband albedo to a surface broadband albedo. Finally, the surface clear sky broadband albedo is adjusted for the effects of cloud cover in cloudy pixels over snow and ice (Key et al., 2001). The reflectance is also corrected for dependencies on sun-satellite-surface viewing geometry. Uncertainties in the retrieval of surface albedo are larger in cloudy sky conditions than in clear sky conditions. Downwelling fluxes at the surface are computed with a neural network, called FluxNet, which is trained to simulate a radiative transfer model (Key and Schweiger, 1998). The neural network uses derived geophysical variables as input (Key and Wang, 2015). To determine the absorbed shortwave energy at the surface, the downwelling shortwave flux was multiplied by the surface absorption (1-albedo) for each pixel. More details of the algorithms are described in Key et al. (2016) and references therein.

The study areas are land and non-land between 60-90°N latitude, where land is typically snow-covered and ocean is ice-covered during the winter, except for parts of the North Atlantic Ocean. Land includes Greenland. "Non-land" is almost exclusively ocean, but does include some inland lakes. For simplicity, we use "ocean" to mean "non-land" throughout the rest of this paper. Over this domain, the land and ocean areas contain a similar number of equal-area pixels, with land areas consisting of 26,682 25 km pixels and ocean areas consisting of 27,674 pixels, a difference in area of 3.58%.

APP-x data show that annual mean absorbed solar radiation at the Arctic surface has increased over the 1982-2015 period (Figure 1). The magnitude of absorption and the rate of increase, however, were different for land and ocean. Trends in surface albedo, surface temperature, cloud cover, and shortwave radiation are calculated using annual mean values with a linear least square fit regression over the thirty-four-year period, and confidence of the trends is calculated using 2-tail student's t test. Over land, the average increase in absorption was 0.21 W m$^{-2}$ year$^{-1}$, significant at the 90% confidence level; Over ocean, the average increase was 0.43 W m$^{-2}$ year$^{-1}$, significant at the 99.9% confidence level. The shortwave absorption increase over ocean was, therefore, approximately two times as large as the increase over land. Absorption over the ocean increased by 0.3% of the annual mean ocean absorption per year, resulting in an approximate 10% increase over 34 years. Over land, the increase was 0.09% of the annual mean per year, or about 2.7% over the study period. The increased absorption over land can be attributed to the decreasing snow cover and hence decreasing albedo, especially in spring (Robinson & Frei, 2000; Déry & Brown, 2007). The increased solar absorption over ocean can be attributed to the shrinking sea ice cover (Pistone et al., 2014; Polyakov et al., 2012). Including or omitting Greenland in the calculations for land has a relatively small impact on the results. If Greenland is excluded, the average annual mean shortwave absorption over land increases by about 18 Wm$^{-2}$ but the strength of the absorption trend is slightly weaker. Greenland's high albedo results in less shortwave absorption than other Arctic land areas, but the decrease in this albedo over time, especially over Greenland's coastal areas, contributed to a stronger absorption trend. Excluding Greenland decreases the absorption trend over land from 0.09% of the annual mean to 0.06%.

The larger trend over ocean than land results from the larger albedo difference between dry, snow-covered sea ice (greater than 0.8) and open water (0.1) (Rösel et al., 2012) than between snow-covered land (0.85) (Greenfell and Perovich, 2004) and land during the melting season (0.2-0.4) (Sturm et al., 2005). Though the change in shortwave absorption over ocean areas outpaces that of land, the greater magnitude of absorption over land, i.e., the actual amount of energy absorbed, is due to greater insolation at lower latitudes. The radiative feedbacks associated with these changes in absorption over both land and ocean are discussed later.

Figure 2 shows the spatial pattern of shortwave absorption trends over the Arctic for April, May, June, and September. These months were chosen because they illustrate the changes during the annual transition from high-to-low snow cover over land (April and May), high-to-low sea ice cover over ocean (June), and the annual sea ice minimum (September). Over Arctic land, the strong increase in absorption due to decreasing springtime snow cover (Robinson & Frei, 2000; Stone et al., 2002) is seen in May. Absorption trends in northern Europe, central Siberia, and the Alaskan Interior are particularly affected by this loss in snow, and this spatial pattern of radiative forcing was also seen by Flanner et al. (2011). Land areas show the greatest absorption increase from March through May, with average May absorption increasing by 1 W m$^{-2}$ per year. Some of the increasing absorption trends in the early spring may be caused by changes in vegetation. Land with more exposed shrub experiences albedo decreases earlier in the year than where there is less shrub or no vegetation at all (Sturm et al.,

2011). Once temperatures are above freezing, sensible heat flux overtakes solar heating and the impact from vegetation causing lower albedo values is reduced (Loranty et al., 2011; Sturm et al., 2005). This means that changes in absorption over snow-covered, vegetative land during the summer months are primarily driven by changes in snow cover, not vegetation. Chapin et al. (2005) determine that on cloud-free summer days, broadband albedo over the Alaskan North Slope has been reduced by 0.0002 per year due to changes in vegetation from 1982-1999.

In contrast, most of the sea ice lasts through early summer, but changes in sea ice thickness and the formation of melt ponds still allow for changes in absorption (Perovich & Polashenski, 2012). Sea ice albedo typically decreases with thickness (Lindsay, 2001), and an increase in melt pond fraction (open water) further reduces surface albedo. As higher temperatures cause the surface of the sea ice (0.8 albedo) to begin melting, the thin layer of water atop the ice (0.6 albedo) can reduce the absolute albedo by 20%. Liquid water more readily absorbs radiation than the surrounding ice and causes more water to pool and create melt ponds, further reducing the ice concentration and albedo of an ice-covered surface (Rösel et al., 2012). Melt ponds that appear early in the melting season allow for greatly increased absorption over sea ice, and may even drive regional-scale sea ice changes in extreme cases (Rösel and Kaleschke, 2012). By late February or early March, sea ice concentration and extent reach their annual maximum under weak sunlight, so absorption trends over the Arctic Ocean are very small. From June to October, however, the multi-decadal changes to the extent, thickness, and the surface albedo of summer sea ice caused the absorption rate to increase faster than absorption over land, particularly in the Beaufort and Chukchi Seas. Flanner et al. (2011) also noted that increases in radiative forcing from 1978-2008 over lower-latitude Arctic seas were greater than those over land during June-October. Sea ice extent and concentration have decreased over the last few decades, and thick, multiyear sea ice that was prevalent in the 1980s and 1990s has lost as much as 50% of its thickness (Kwok & Rothrock, 2009), if not vanished altogether (Serreze et al., 2007). First year ice is more susceptible to the formation of melt ponds, which can cause precipitous decreases in albedo (Rösel et al., 2012). The increase in surface absorption over the Arctic Ocean, then, is due to a combination of the replacement of multiyear sea ice with first year ice and open water over the study period.

While the increase in the absorption of shortwave radiation is largely due to reductions in sea ice and snow cover extents, the linear correlations between snow cover anomalies or sea ice extent anomalies and shortwave absorption anomalies are both approximately -0.6 (not shown). Regional and seasonal changes in cloud cover explain some of the variance in these relationships. The 34-year trends in cloud cover were explored using APP-x data from 1982-2015. Over land, an increase (decrease) in highly reflective cloud cover is associated with decreases (increases) in surface absorption. For example, Arctic land areas that have experienced an increase in cloud cover (Alaska, western Russia, and north central Siberia) show decreasing trends in shortwave absorption. The spatial variability of the surface shortwave absorption over land in Figure 2 can be explained, in part, by trends in cloud cover. Figure 3 provides an example for September, where both positive and negative trends in cloud cover over eastern Siberia show a strong relationship with trends in absorbed solar radiation. While

portions of the Arctic Ocean have also experienced changes in cloud cover, their effect on trends in shortwave absorption are much less, primarily because most of the ocean is still ice-covered and the reflectivities of ice and cloud are similar. We found that the trends in absorbed shortwave radiation over land are more affected by changes in cloud cover than over the ocean, and that trends in cloud cover can result in radiative absorption increases or decreases over land during the period of study, as shown in Figure 3.

While it can be seen qualitatively that the regional effect of clouds can be large, quantitatively determining their overall influence on the trend in absorbed shortwave radiation, i.e., to separate the influence of changes in cloud cover from changes in sea ice and snow cover, is not possible with the data available. Instead, we quantify the contribution of clouds by determining their maximum possible effect on downwelling shortwave radiation at the surface over the study period. This is done by using the 34-year average downwelling shortwave surface flux for each of the sunlit months (March-September) and the 34-year average cloud cover trend (fractional cloud cover) to determine the changes in instantaneous surface shortwave flux. At each grid point in each month, the 34-year average downwelling flux is multiplied by the cloud cover trend. A positive cloud cover trend will result in a decrease in the downwelling and vice versa. For this calculation it is assumed that all clouds are optically thick ("black") and reflect almost all incident sunlight, as optically thick clouds would have the maximum effect on downwelling shortwave radiation. This assumption is valid based on Wang and Key (1995), which finds that visible optical depths for Arctic clouds are in the range of 5-6, corresponding to a transmittance of near zero (0.2-0.6%). The average cloud cover trend over land and ocean are -0.265% and -0.392%, respectively, or 10% and 4% of the change in shortwave absorption. For March-September, changes in cloud cover from 1982-2015 resulted in an increase of surface absorption by 1.94 $Wm^{-2}$ over land and 2.19 $Wm^{-2}$ over ocean. These cloud-based changes in surface absorption account for only a 0.5% increase in surface insolation over the ocean and a 0.4% increase over land.

Even though September experienced the greatest decrease in sea ice extent, the smaller incoming solar flux at this time of year result in smaller absorption increases than those of early summer. The early spring, late fall, and winter months exhibit far weaker trends in shortwave absorption over ocean than land due to lower variability in the sea ice cover and smaller solar fluxes - decreasing to zero in the winter - at the high latitudes.

Surface radiation and cloud cover data from the National Aeronautics and Space Administration (NASA) Modern-Era Retrospective Analysis for Research and Applications 2 (MERRA2) reanalysis (Rienecker et al., 2011) are employed to provide verification of the results from APP-x. This study used MERRA2 version 1.3 and determined the absorbed shortwave radiation trends at the surface from the surface incoming shortwave flux (SWGDN) surface albedo (ALBEDO) variables.

Performing the same analysis as before on MERRA2 data produced similar results. The trends in absorbed radiation for the month of June from APP-x and MERRA2 show similar patterns, though with larger magnitudes in APP-x (Figure 4). The reanalysis data show an increase in absorption over ocean during June and mixed trends over land, which correspond spatially to APP-x trends. The results were consistent with APP-x, with increasing, uniform ocean heating during high summer, and changes over land influenced by factors other than surface albedo. The most obvious differences between the reanalysis data and APP-x occur over the central Arctic Ocean, where MERRA2 absorption trends are weaker than those in APP-x. The cause of this difference is due to the fixed albedo value that MERRA2 assigns to sea ice, which does not take sea ice thickness or melt ponds into account. As seen in the APP-x results, thinner ice and irregularities in the ice surfaces increase the absorbed surface radiation.

## 3 Timing of Transition from High to Low Albedo

The trends in solar energy absorption at the surface are both a result of, and a forcing for, changes in surface albedo. As increasing solar absorption over the Arctic continues to affect land and ocean differently, we now explore how the timing of the low albedo portion of the year has changed over time, and how the timing relates to the available solar energy. Markus et al. (2009) determined that between 1979 and 2007, nearly all regions of the Arctic showed a trend towards earlier annual melting and later refreezing, which self-enhances as sea ice thickness decreases. Results presented here are consistent with their analysis and expand upon the surface energy implications.

Using APP-x data, we are able to track the changes of land and ocean albedo throughout the study period. The impacts on the surface energy budget are apparent in Figure 2. However, the absolute timing of the low albedo period as well as the shift in timing of this period over the last few decades require further examination. One approach to analyzing these changes in the land and ocean albedo is to determine the day-of-year (DOY) in which the average albedo over land and over ocean reached their minima for each year. However, due to late freezing and thawing events and dynamically-driven changes in the sea ice edge, changes in the albedo minimum DOY do not accurately explain trends in absorbed solar energy over the last 34 years. We find that using the DOY range from when the Arctic transitioned from a relatively low albedo (the day that albedo first went below 0.4) to a very-low albedo (the day that albedo went below 0.25) state provides a better metric for comparing the changes in albedo over land and ocean (Figure 5). Figure 5 shows that the majority of the snow cover over land melts earlier in the year than sea ice, which is due to higher sun and temperatures at a lower latitude. Terrestrial snow cover also melts earlier because the snow-free land adjacent to snow-covered land warms faster than the unfrozen ocean around the sea ice.

An examination of Figure 5 shows that Arctic has reached a lower albedo state increasingly early in the calendar year over both land and ocean since 1982. A linear fit of the midpoint between the days of year at which the 0.4 and 0.25 albedo levels

were reached shows a decrease of 0.64 days per year over ocean and 0.62 days per year over land over the last 34 years. Both of these trends are significant at the 99.9% confidence level. Furthermore, the rate at which the albedo is decreasing from 0.4 to 0.25 has accelerated. Over ocean, a linear fit of the length of the interval showed that it took over 16 days for the average ocean albedo to decay to 0.25 from 0.4 in the initial years of the study. By the end of the record, this albedo decrease took just 8 days (significant at the 99.9% confidence level). Over land, the change in rate of albedo decrease showed a prominent decrease from 17 days to 9 days over 34 years, although the statistical confidence level is less than 90%.

The regression in time of the low-albedo period towards earlier in the year over both land and ocean may have important radiative implications in the future. Over ocean, the low-albedo period was reached two weeks closer to the summer solstice (DOY 172) in 2015 than in 1982-1985, with the low-albedo range midpoint going from DOY 188 to DOY 167. Over land, the low-albedo range midpoint regressed nearly 20 days away from the summer solstice, closer to DOY 152. Though both land and sea experienced lower albedos migrating closer to the beginning of the year, the low-albedo period over land now occurs before the summer solstice, while the low-albedo period over the ocean occurs closer to the solstice and therefore at a time with much greater solar insolation. Even though the insolation during the low-albedo period is greater today than it was in the early portion of the study, the midpoint of the low-albedo interval has regressed past the summer solstice in the last few years (Figure 5). This implies that current trends in sea ice changes may cause the albedo transition to occur even further towards the beginning of the year, thereby experiencing weaker insolation, similar to the regression of the low-albedo period over land. As such, the differences between land and ocean absorbed shortwave radiation trends may grow smaller as their albedo transition occurs earlier in the year.

The magnitude of insolation on any given day at the peak solar time is greater at lower latitudes. Therefore, even small changes in albedo in the lower Arctic can have large effects on the amount of energy absorbed at the surface. Conversely, large changes in albedo at higher latitudes are required to significantly affect shortwave absorption due to the weaker instantaneous insolation at higher latitudes. For instance, in 1982, the average albedo of all ocean pixels at 75°N was 0.345 on July 1. By 2015, the average ocean albedo on that date had decreased to 0.234, a change of over 11% (absolute). The corresponding change in average absorbed shortwave energy at 75°N on July 1 between 1982 and 2015 was 14.3 W m$^{-2}$. In contrast, the average land albedo at 65°N on July 1 decreased only 1.6% (absolute) between 1982 and 2015, yet the change in absorbed energy over land (4.8 W m$^{-2}$) was 34% of the change that occurred over ocean. At 75°N, albedo must decrease three times as much as it does at 65°N for the same increase in absorption in July, based on differences in the magnitude of insolation.

However, the magnitude of the flux accumulated over the entire day around the summer solstice is larger at higher latitudes. Figure 6 provides a simple illustration of the changes in the accumulated, top-of-atmosphere, incoming shortwave flux at 65°N and at 75°N. For an equivalent change in albedo, the accumulated absorbed TOA shortwave flux is larger at

higher latitudes because, even though the sun is lower, there are more hours of sunlight. The change in TOA absorbed accumulated flux at 65°N is 96% of the change in accumulated flux at 75°N. This relationship is also true at the surface. The accumulated flux on July 1 was calculated for the average ocean and land surface at 65°N and 75°N using albedos from the years 1982 and 2015. Results showed that at both latitudes, the accumulated flux on July 1 increased more over ocean

5 between 1982 and 2015 than over land. Accumulated flux increases over ocean at 75°N (4.73 MJ) were more than twice as high as the changes at 65°N (2.05 MJ). Accumulated flux changes over land at 75°N (3.11 MJ) were also much higher than at 65°N (0.16 MJ). The greater changes in accumulated flux are related to larger albedo decreases at higher latitudes, where snow cover and sea ice may have changed more drastically than at 65°N. Therefore, at both latitudes over the last 34 years, the average ocean pixel has experienced a greater change than the average land pixel.

Figure 6 also shows the changes that occur due to a low-albedo regression towards earlier times of the year. Over ocean, the shift in the timing of lower albedos to earlier in the year means that more sunlight was absorbed over the ocean in 2015 than in 1982, all else being equal (e.g., cloud cover). Over land, the regression of low albedo towards earlier in the year still results in an increase in absorbed energy, but it can only increase modestly due to decreasing sunlight further from the

15 summer solstice. This relationship is valid for both the peak solar time and the accumulated absorbed fluxes.

## 4 Stronger Snow-Albedo and Ice-Albedo Feedbacks

The increased solar absorption due to the temporal regression of the low-albedo period results in a positive surface albedo feedback. One way to define the strength of the albedo feedback is the change in net incoming shortwave radiation with respect to surface temperature due to changes in surface albedo (Cess and Potter, 1988; Qu and Hall, 2007; Fernandes et al.,

2009):

$$\frac{\partial Q}{\partial T} = -I \frac{\partial \alpha_p}{\partial \alpha_s} \frac{d\alpha_s}{dT}$$

where $Q$ is the net (absorbed) shortwave radiation at the top of the atmosphere (W m$^{-2}$), $I$ is incoming solar radiation at the TOA surface (W m$^{-2}$), $T$ is temperature (K or C), $\alpha_p$ is the planetary albedo at TOA, and $\alpha_s$ is the surface albedo. The term $I$ is calculated as the montly mean incoming solar radiation at the APP-x grid level. The term $\partial\alpha_p/\partial\alpha_s$ over land and over ocean is calculated using an analytical model developed by Qu and Hall (2007) and surface albedo for all-sky and clear-sky only,

albedo at TOA for all-sky and clear-sky only, cloud amount, and cloud optical thickness monthly means from APP-x 1982 to 2015 at the APP-x grid level. Coefficients required in this analytical model, $\varepsilon_1$ and $\varepsilon_2$ in each month, are derived following Eq.10 in Qu and Hall (2007) with collocated monthly means of planetary albedo at TOA for all-sky and clear-sky, cloud amount, cloud optical depth, and surface albedo at the APP-x grid level as a regression sample; another parameter (coefficient) required in this model is $T_a^{cr}$, effective clear-sky atmospheric transmissivity, which is derived monthly

following Eq.5 in Qu and Hall (2007), using each collocated planetary albedo at TOA for clear-sky and surface albedo as a regression sample. The term $I\,\partial\alpha_p/\partial\alpha_s$ over land and over ocean is calculated following Eq. 12 in Qu and Hall (2007) at the APP-x grid level and then averaged. The term $d\alpha_s/dT$ over land and over ocean is calculated as the averaged ratio of the monthly surface albedo trend to the monthly surface temperature trend at the APP-x grid level following Fernandes et al. (2009). All procedures follow Qu and Hall (2007) and Fernandes et al. (2009).

For a unit temperature change, the net solar radiation absorbed by the earth system over ocean is less than that over land in April, but about four times as large as that over land in June and July (Figure 7). The feedback strengths in June are 16.3 W m$^{-2}$ K$^{-1}$ over ocean and 3.8 W m$^{-2}$ K$^{-1}$ over land. The stronger surface albedo feedback over the ocean at the high-sun time of the year will amplify the warming effect, allowing for even more solar radiation to be absorbed by the earth system in the future, pushing the low-albedo threshold back even earlier in the year, and leading to a further decline in the Arctic sea ice cover.

## 5 Conclusion

The surface radiation budget of the Arctic is strongly influenced by changes in albedo, cloud cover, moisture, and heat advection. This study examined multi-decadal changes in the amount of solar radiation absorbed at the surface of Arctic land and ocean, together and separately, as a result of changes in albedo due to decreasing sea ice and snow cover. Analyses of the APP-x satellite dataset and the NASA MERRA2 reanalysis over the 34-year period 1982-2015 determined that the magnitude of shortwave absorption is greater over land than the ocean, and that changes in snow and sea ice cover have led to an increase in absorbed shortwave radiation of 10% over ocean and 2.7% over land. It was found that the rate of change in absorption over the Arctic Ocean is more than double the rate over Arctic land, and that the magnitude of the ice-albedo feedback is four times that of the snow-albedo feedback in summer. However, the difference in the trend in shortwave absorption between land and ocean may decrease as the low-albedo period occurs further away from the summer solstice.

The timing of the annual low-albedo period has changed and has changed differently for land and ocean. While similar studies assume a consistent albedo cycle when determining the cryosphere's contribution to the global energy budget (Flanner et al., 2011), here we find that the inclusion of inter-annual changes to surface albedo result in a significant change to the surface shortwave energy budget of the Arctic between 1982 and 2015. Since 2010, for example, average ocean albedo in the study area during late June has been as low as mid-September albedo in 1982-1985. Similarly, Arctic land is losing its snow cover earlier in the year. If these trends continue, the temporal regression of the low-albedo period over land and ocean will have different effects on absorbed solar radiation in the future because the low-albedo period has moved further from the high-sun/maximum insolation time of year over land, but has moved closer to the high-sun time over ocean. This has

resulted in an intensification of the ice-albedo feedback more than the snow-albedo feedback, which may decrease as snow and ice melt earlier in the year. The absorption changes illustrate the relative importance of the snow-albedo feedback and the ice-albedo feedback, and point toward the decreasing sea ice cover, not changes in terrestrial snow cover, as the foremost radiative feedback mechanism affecting recent and likely near-future Arctic climate change.

**Acknowledgements**

This research was supported by the NOAA Climate Data Records program and the Joint Polar Satellite System (JPSS) Program Office. The views, opinions, and findings contained in this report are those of the authors and should not be construed as an official National Oceanic and Atmospheric Administration or U.S. Government position, policy, or decision. We thank the anonymous reviewers and the editor for their valuable comments and suggestions.

**Author Contribution**

All authors contributed to the writing of this paper. Aaron Letterly performed much of the data analysis and drafted the manuscript. Yinghui Liu led the snow-albedo and ice-albedo feedback section. Jeffrey Key formulated the research idea and goals, and performed some calculations.

**Competing Interests**

The authors declare that they have no conflict of interest.

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

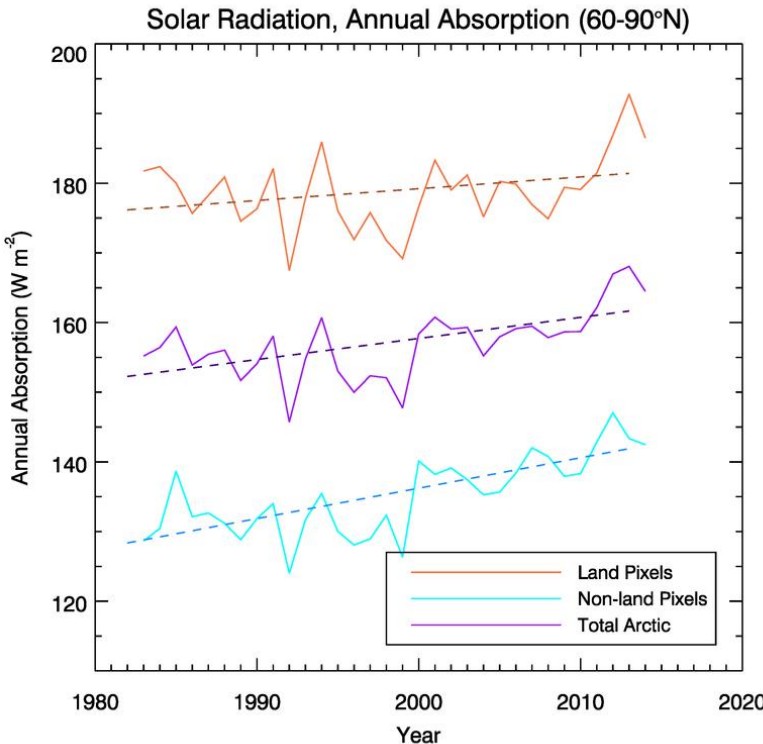

**Figure 1: Average annual surface shortwave absorption (W m$^{-2}$) from 60-90°N for the combined land and ocean area (purple), land only (orange), and ocean only (cyan). Dotted lines are linear trends.**

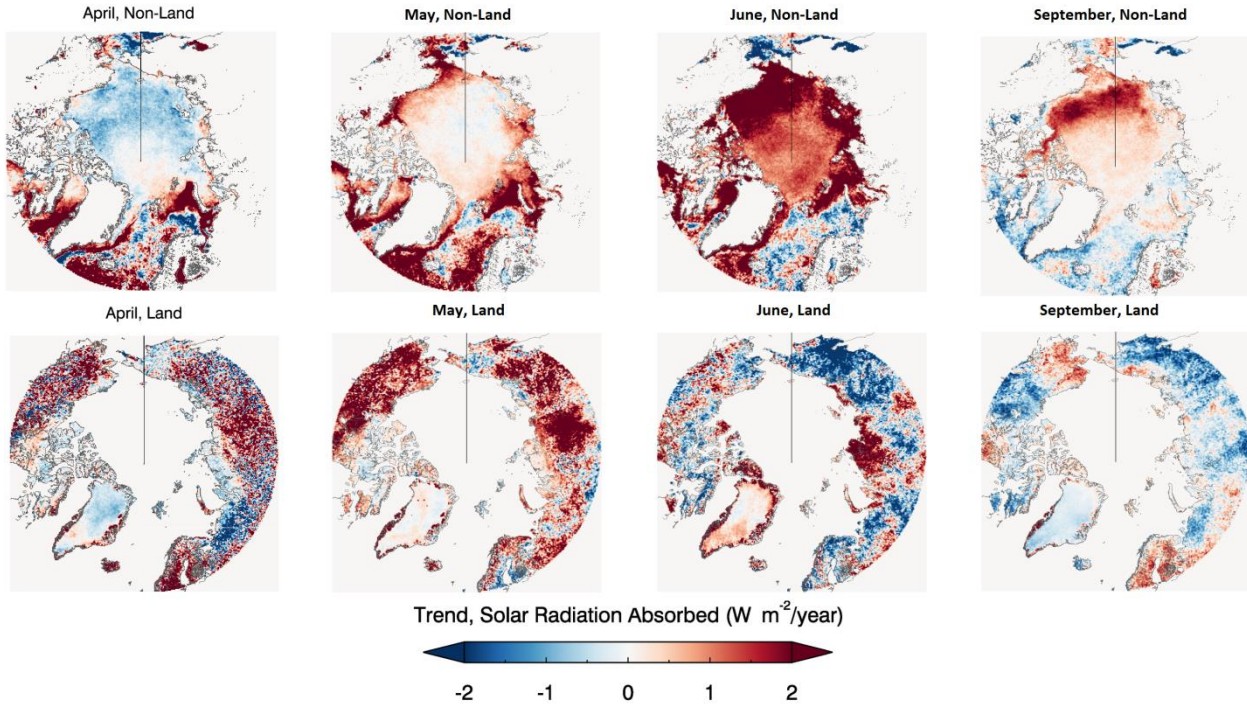

**Figure 2: Trends in absorbed radiation for selected months over ocean (top row) and land (bottom row).**

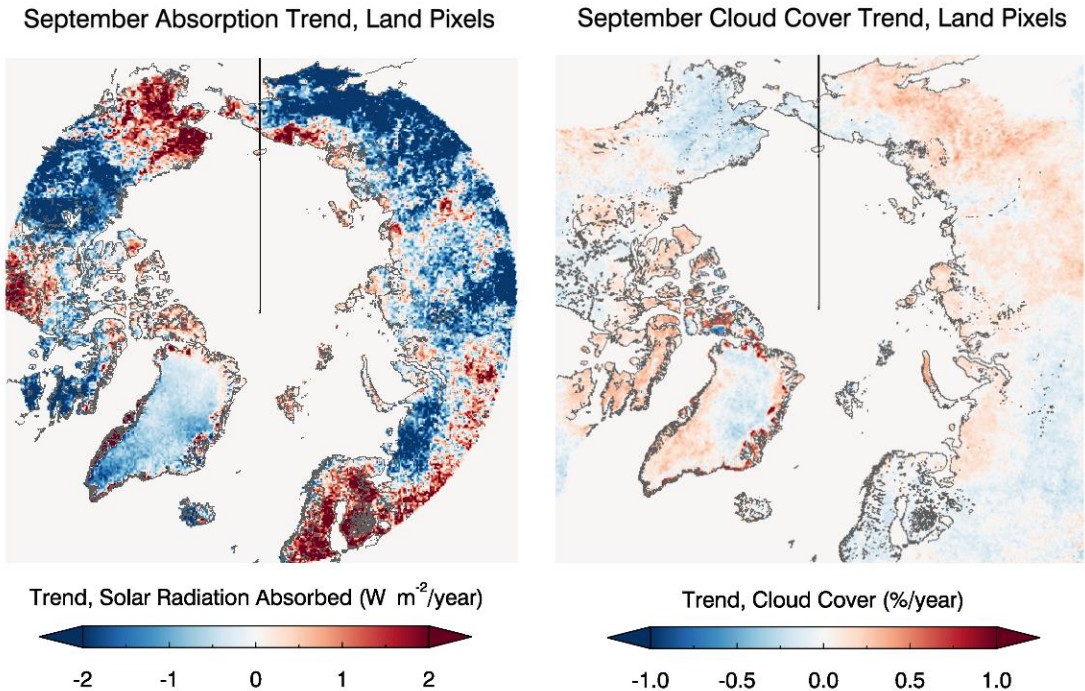

Figure 3: Trends in absorbed shortwave radiation over land (left) and cloud cover trends over land (right) during September.

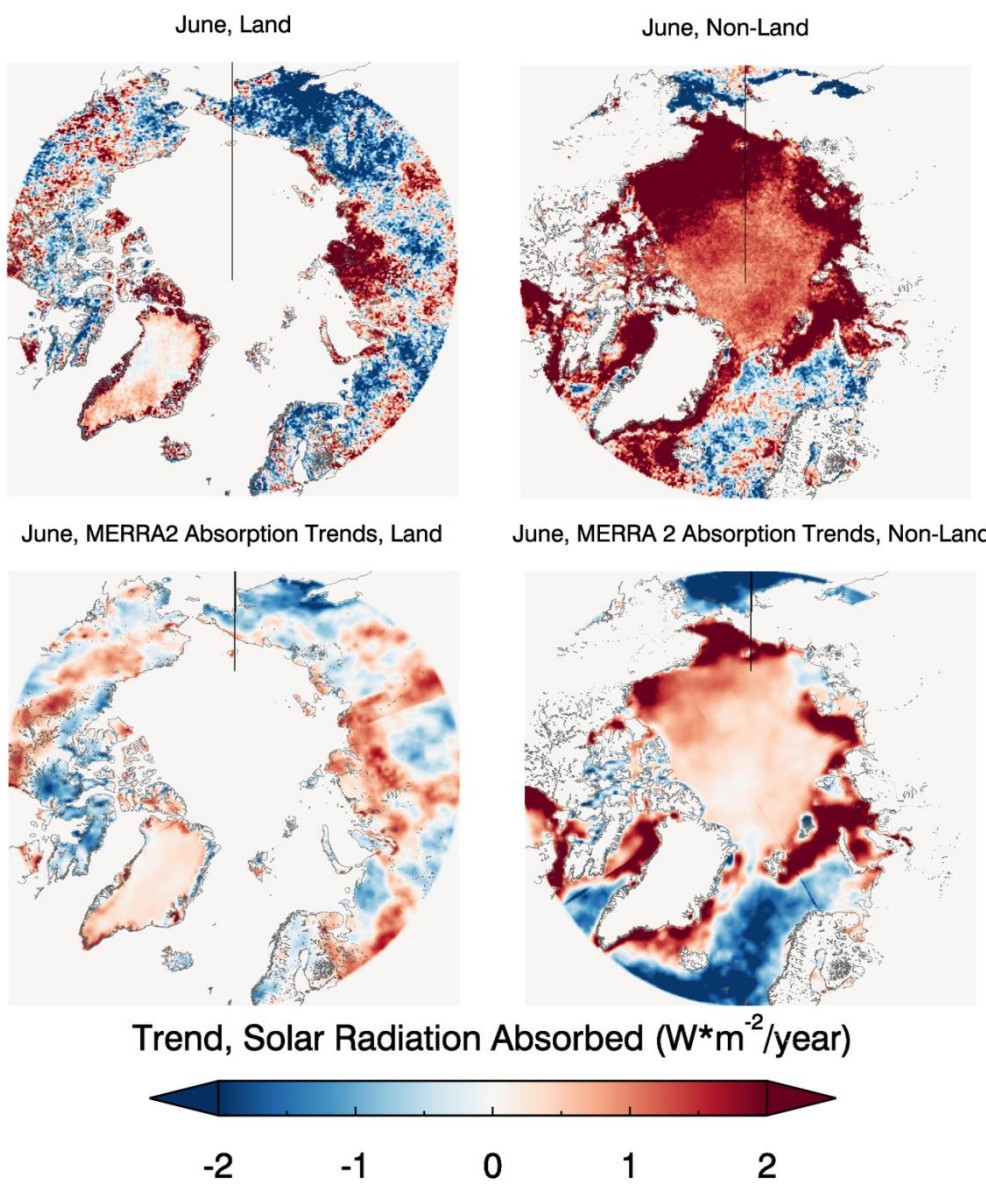

**Figure 4: Trends in absorbed radiation from APP-x over land (top left) and ocean (top right) compared to trends from MERRA2 over land (bottom left) and ocean (bottom right) during June.**

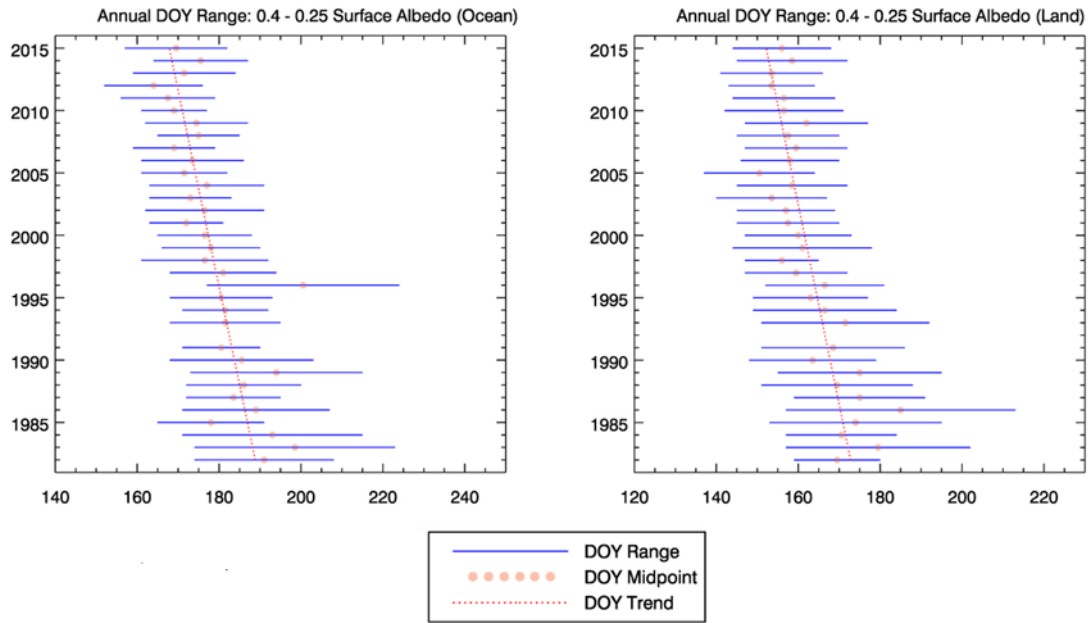

**Figure 5: Day of year range between 0.4 and 0.25 albedo over ocean and land (blue) from 1982 to 2015. The dotted trend line (red) shows the regression of the DOY midpoint (pink) over the time period.**

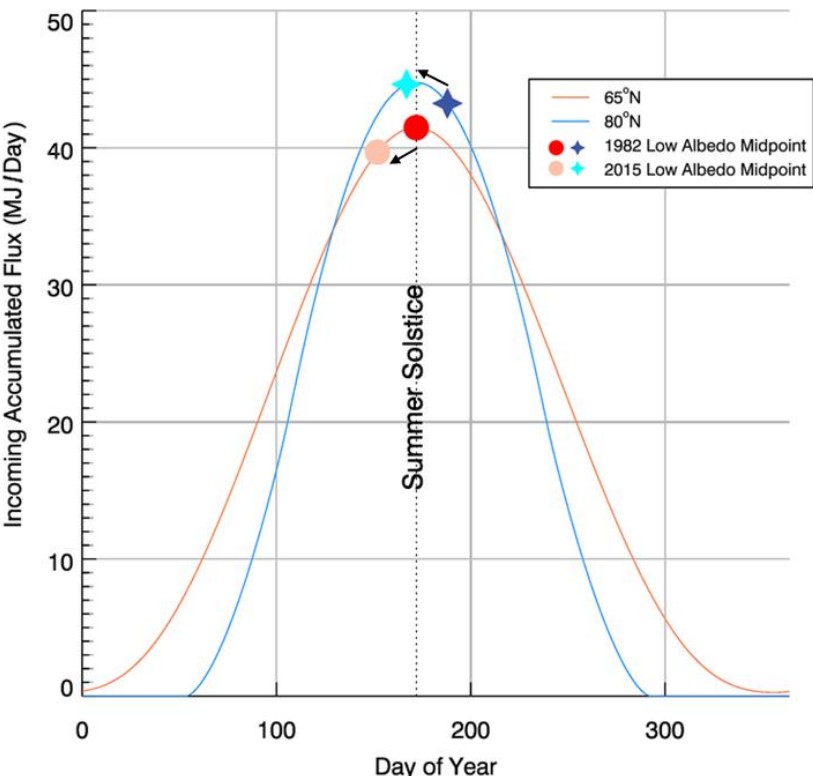

**Figure 6: Accumulated top-of-atmosphere incoming shortwave flux for each day and for the 65°N (orange) and 80°N (blue) latitudinal bands, roughly representing the Arctic Ocean and Arctic land, respectively. Darker symbols represent the day of year that the midpoint trend of the low-albedo period (Figure 5) was reached over land (circle) and ocean (star) in 1982, while lighter symbols show the day of year of the 2015 low-albedo period midpoint trend. Arrows clarify the direction in time for the change in the low-albedo period midpoints.**

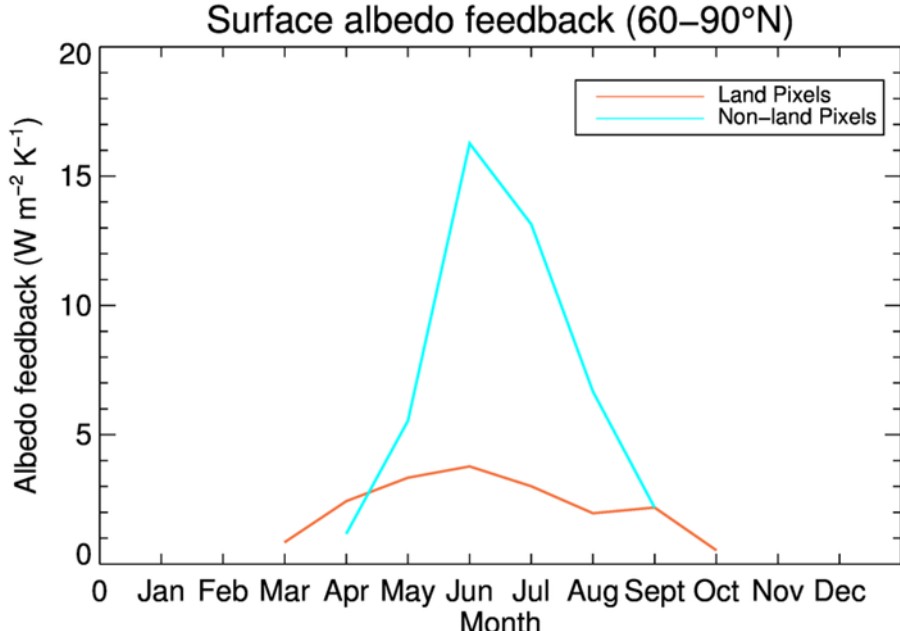

**Figure 7: The snow-albedo and ice-albedo feedbacks (equation 1) for Arctic land (orange) and ocean (cyan) for the period 1982-2015.**