# Peer review of "Arctic Climate: Changes in Sea Ice Extent Outweigh Changes in Snow Cover"

_The Cryosphere, 2018_

## Referee Comment (RC1) · Anonymous Referee #1 · 16 Jul 2018

This study explores and compares the contributions of recent (satellite-era) changes in Arctic sea ice and snow cover to changes in absorbed solar energy at the surface, and finds that sea ice losses have contributed to a greater increase in solar heating than reduced Arctic snow cover. Similar analyses have been performed before, though this study carves out a niche by focusing exclusively on the Arctic (60-90 degrees) and comparing the terrestrial and sea ice contributions within this domain. An important point highlighted by the authors is that the major seasonal transition in albedo associated with sea ice loss is occurring near the summer solstice, whereas the terrestrial snow transition is shifting away from the solstice, implying greater solar forcing via sea ice loss in the recent past and near future. Overall, the paper is concise and very well-written, though there are several important aspects of the analysis that need to be

revisited and/or clarified before publication. After these issues (described below) are addressed, I would support publication of this manuscript in The Cryosphere.

Major issues:

(1) The discussion on p.7 lines 4-14 describes how lower-latitude changes in albedo drive larger changes in absorbed solar energy than equivalent albedo changes at higher latitudes. This is true for annual-mean albedo changes, but it is not true for the summer-solstice-season changes that are the focus of this section. This can be seen clearly in the authors' own Figure 5, which shows greater daily-mean solstice insolation at 80N than at 65N. The discussion on p.7 lines 4-14 is therefore largely inconsistent with Figure 5 (the latter of which is correct, I believe). The authors need to amend the discussion and re-consider potential causes of the statistics described on p.7. Differences in cloudiness and cloud trends may be a logical starting point for resolving this discrepancy.

(2) The study focuses largely on changes in net shortwave flux at the surface, and the authors acknowledge at different points in the text that changes in clouds and sea ice thickness could contribute to net shortwave changes, in addition to the more obvious contributions of changing snow and sea ice coverage. Section 2 could be improved with a bit more quantification of how large these other contributions are. Perhaps such quantification is beyond the scope of the study, though I encourage the authors to con- sider ways in which they could quantify the contributions of these different drivers of shortwave flux trends. It seems that the cloud contribution could be isolated and quan- tified via existing APP-x data, though isolating the influence of sea ice thickness/age would be more challenging.

Related to the above point, the discussion on the top of p.4 highlights a trend of lower albedo over Greenland's near-coastal regions. Is this trend caused more by reduced snow cover over the (quite small) non-glaciated portion of Greenland, or more by dark- ening of the perennial ice surface?

(3) A companion analysis of MERRA2 data is referred to very briefly at the end of section 2. This is a nice addition to the study, but it would be helpful if this analysis was developed more. In particular, it would be helpful to state the shortwave trends obtained from MERRA2 and show companion figures to Figs 1 and 2, so that similarities and differences between the two products can be seen more clearly. Presenting results from two or more products will give the study more credence.

(4) Figure 5 shows that the "Low Albedo midpoint" over oceans actually occurred *before* the solstice in 2015. If this is now the norm, it implies that future trends towards earlier melt will, as with terrestrial snow, cause the high-low albedo transition to move away from the solstice. In other words, the lag between snow and sea ice melt, in combination with melt trends, may have caused the snow/sea ice forcing differences to have *already* peaked. In light of this (and if I have interpreted correctly), the authors may want to add a bit of nuance to the abstract and associated discussion.

(5) The analysis of feedbacks in section 4 should either be removed, or methodological details of this analysis need to be clarified. Personally, I think this section could simply be removed, as it does not add much to the study, and is likely sensitive to methodological choices. If it is retained, more detail is needed on methodology, including how the spatial and temporal averaging of temperature (in particular) was conducted. Were monthly or annual trends in T used? Were gridcell-level or Arctic (or global) T trends used? (e.g., how exactly were the presented June feedback numbers calculated?) Personally, I think feedback analyses are only meaningful when large areas (e.g., hemispheric or global) are used for temperature averaging. More detail would also be needed on the technique used to determine: $(d\ alpha\_p\ /\ d\ alpha\_s)$.

(6) Conclusions, p8,13-17: This passage could be important, but needs clarification. To be clear, Flanner et al (2011) assumed constant seasonal cycles of the albedo of multi-year and first-year sea ice. Thus interannual changes in sea ice extent and transitions from multi-year to first-year ice did contribute to area-averaged albedo changes in that study, but changes in ice albedo due to thinning or earlier ponding (etc) did not. p.8

line 15 states that "...here we find that the inclusion of inter-annual changes to surface albedo result in a significant change to the surface shortwave energy budget...". Do the "inter-annual changes to surface albedo" refer to changes in the albedo of the ice itself? If so, I do not recall seeing quantification of this contribution in the analysis (though I think it would be very useful!). The subsequent sentence also needs clarification: "Since 2010, for example, average ocean albedo in the study area during late June has been as low as mid-September albedo in 1982-1985." Are you referring to the ocean-wide albedo, or to the albedo of the sea ice itself? If the former, this is likely due simply to the reduction in June sea ice extent, and this would have been accounted for in the Flanner et al study. If not, this is a useful finding, but one that should be reported and further developed earlier in the study.

Minor comments:

General: I appreciate the focus of this study on the "Arctic", defined here as 60-90N, but it is important to note that any conclusion about the relative magnitudes of sea ice vs. snow changes will be sensitive to the latitude bands selected. For example, if the latitude threshold for the analysis was adjusted equatorward, the relative contribution of land snow changes would clearly increase. I think this point should be acknowledged more clearly.

p1,21: A slightly clearer way to say this would be "September sea ice extent decreased by 45% from..."

p2,30: 1972 should be 1979, in reference to the Flanner et al study.

p6,22: This either needs "although" after the comma, or it should be two sentences.

p6,24-30: Please list the solstice DOY, for reference.

Figure 1: It looks like there is only one point per year in this figure, so I assume you mean "Annual Absorption" instead of "Monthly Absorption".

Figure 2: Could you speculate on the cause of the negative trend in April non-land

albedo?

---

## Referee Comment (RC2) · Anonymous Referee #2 · 3 Aug 2018

Background

The authors use 34 years of satellite data from AVHRR (1982 – 2015), over the Arctic to study the impact of decreases in ice and snow on the trend of solar absorption in that region. They find large differences in the absorbed solar energy over the Arctic Ocean as compared to the land areas and estimate that the magnitude of the ice-albedo feedback is four times that of the snow-albedo feedback in summer.

General comments

1. It is stated that the time of the high-to-low albedo transition each year is moving toward the high sun of the summer solstice over ocean but moving away from the summer solstice over land. Some explanation is needed why this happens.

[Figure]

2. They claim that decreasing sea ice cover, not changes in terrestrial snow cover may play an even larger role in future Arctic climate change. The paper is not about prediction of future state of the arctic so there is no substantiation of what might happen in the future.

3. There is confusion when dealing with the surface and top of the atmosphere (TOA). At TOA not only surface propertied matter but also clouds so it is a mixed signal (example to be given in Specific Comments).

4. There is some lack of clarity about the impact of changes at the surface and the latitudinal changes in the solar radiation reaching the ground on the amount of absorption at the surface. How do you separate these two factors?

5. MERRA-2 is also used in the analysis. Nothing is said about the differences in spatial and temporal scales of MERRA-2 and AVHRR. What impact does it have on the conclusions?

Specific Comments

Stated: Between 1979 and 2011, the Arctic top-of-atmosphere (planetary) albedo decreased from 0.52 to 0.48 (Pistone et al., 2014), and subsequent years with record or near-record low sea ice extent have further increased the amount of heat absorbed in the Arctic (Pistone et al., 2014). Comment The connection here between TOA and surface is confusing. Stated: Snow extent has decreased over Eurasia and North America since the late 1980s Robinson & snow cover and the radiative balance over mid- and high-latitude land in the Northern Hemisphere (Groisman et al., 1994), in which retreating snow cover has led to a lower polar Comment The topic is the Arctic, so need to be focused. Stated: preconditioning of sea ice in the winter can influence the albedo into the fall of the following year, illustrating how changes in cloud cover during different seasons may affect the planetary albedo (Letterly et al., 2016; Liu & Key, 2014). Comment Above is a mixed bag of statements. Needs to be cleared. Stated: This study focuses on the effects of snow and ice cover changes on the surface shortwave radiation budget of the Arctic - defined as the area poleward of 60°N - not the remote effects of mid-latitudes on the Arctic Comment Why to invoke remote effects of mid-latitudes on the Arctic? This topic is not of relevance here and the comment does not add much to the discussion. Stated: Furthermore, since terrestrial snow cover has mostly melted by June, the main drivers of absorption trends over land during the summer may be changes in cloud cover or vegetation (Chapin et al., 2005; Loranty et al., 2011)

Comment There is no discussion of changes in vegetation after melt and its impact on absorption.

Stated: In contrast, most of the sea ice lasts through early summer, but changes in sea ice thickness still allow for changes in absorption (Perovich & Polashenski, 2012). Comment The impact of ice thickness on absorption was not addressed in this paper. Needs more discussion.

In summary, additional work is needed to streamline the text, add explanations and remove redundancy.

---

## Referee Comment (RC3) · Anonymous Referee #3 · 7 Aug 2018

General comments:

This study examines a dataset (APP-x) that has not been considered in such detailed analysis in the past. The authors examine spatio-temporal trends in absorbed short-wave energy (and other parameters) for a time period (1982-2015) during which the Arctic land-ice-ocean system underwent major changes. The study is an original and timely contribution that is of particular value because it compares changes over land with those over the oceans, and evaluates the magnitude of trends (and indirectly, feedbacks) in detail. The paper is brief and conveys a number of key points in a small space, which is both a strength and a weakness. In regards to the latter, some of the discussion of underlying causes is too simplistic at best. Other parts of the paper would also benefit from a more in-depth, nuanced discussion. Both of these issues

are discussed in more detail below. Overall, the paper is an important and significant contribution.

Specific comments:

(1) The analysis and discussion of albedo differences and trends over sea ice (beginning on p. 4, bottom) is much too simplistic and misleading, since it implies that seasonal evolution and inter annual trends are driven entirely by changes in ice thickness. However, prior to the onset of melt the differences in snow albedo on different ice classes (with the exception of very thin ice) are likely insignificant. Much more important in this context are development, areal fraction, and optical properties of ponds on sea ice. It is here that major contrasts between different ice age (MY vs. FY ice) and ice thickness are expressed. Moreover, these processes are dominant for the months of June and July, such that the discussion of Fig. 2 (June) in terms of ice thickness classes is not really appropriate. Here, a more rigorous discussion of the observed trends in terms of the seasonal cycle of ponds on sea ice (Perovich and Polashenski, 2012; Polashenski et al., 2012; Rösel and Kaleschke, 2012a) is needed. In particular, the work by Rösel and Kaleschke (2012a,b) is highly relevant because it discusses the role of spatio-temporal variations in ponding on sea ice in the context of sea ice concentration and extent anomalies based on remote sensing data. A closer examination of their findings may help explain some of the spatial patterns seen in Fig. 2 and the inter annual variations shown in Fig. 1. It is also relevant in the discussion of reductions in albedo in the month of June (p. 4, l. 18) which is as much or more a function of ponding as of reduction in ice concentration.

(2) The authors attribute changes in albedo over land to changes in snow cover duration during the snow-covered period, and to changes in vegetation and cloudiness after loss of snow cover. This may be too simplistic and requires further analysis. First, while the albedo contrasts between clouds and snow cover may not be as large as those between land surface and clouds, they cannot be ruled out as important without further analysis. For example, the spatially coherent trend for the month of April

towards reduction in absorption of shortwave energy in NW Siberia may well be due to changes in cloudiness rather than duration of snow cover (which persists well beyond April). Without specific references to the published literature or some additional analysis of a particular subregion in a case study it is difficult to accept the explanation offered by the authors wholesale. Second, some of the discussion of snow and ice albedo variations needs to be reviewed and potentially revised. For example, on p. 4, l, 11ff a difference between snow albedo over sea ice (0.6) and land (0.7) is seen as being important. Where do these estimates come from? The paper by Sturm et al. cited here only discusses tundra snow. Both values are low if they refer to dry early spring snow before the onset of melt. Moreover, I am not aware of data showing that the albedo of snow covered sea ice is that much lower than that of tundra snow. This needs to be either corrected or further substantiated.

(3) The discussion of the spatial patterns of trends could be expanded a bit, ideally by referencing either published work or at least a slightly more in depth analysis for a subregion. It is asserted that spatially coherent trends such as that in Greenland or Siberia are driven by trends in cloudiness. This appears plausible, but would benefit from some more detail. However, this raises the question as to what spatially heterogeneous trends are driven by (e.g., April over much of the landmass, or June in much of North America). Are the trends for individual grid cells or small aggregations of grid cells significant if they are neighboring on grid cells with opposite sign in the trend?

(4) The previous comment relates to a significant shortcoming in the manuscript that should be easily remedied. Specifically, the discussion of the methods employed in deriving the different data sets and their analysis is currently much too superficial. First, it would be preferable to separate the description of the datasets used and the analysis methods employed from the reporting of results. Specifically, l. 24 on p. 3 would be a natural break. Then, while reference to the paper by Key et al. (2016) to describe the data product is fine, the current paper needs to provide more information on how the data sets were generated in particular as relevant to the specific variables

(albedo and absorbed shortwave energy, for example) discussed here. For example, how has broadband albedo been derived from spectral radiances and what are the associated errors and uncertainties associated with such a derivation? With the ocean regions located at higher latitudes than the land areas, does this introduce a potential bias because of lower solar zenith angles relative to sensor zenith angles? With a lack of bireflectance distribution function (BRDF) data over melting sea ice as opposed to snow cover over land this may introduce significant uncertainties as well. Further details for the MERRA reanalysis products should also be provided in a restructured methods and data section.

Specific edits: - p. 1, l. 4: For clarity throughout it may be better to refer to sea ice albedo feedback and snow albedo feedback (or at least clarify at the start that ice albedo feedback only refers to sea ice) - p. 1, l. 12: reads a bit awkward, maybe change to something like "the timing of the seasonal transition from high to low albedo ... shifting towards greater insolation associated with summer solstice" - p. 3, l. 27: change to "confidence of the trends is calculated" - p. 4, l. 2: Not sure how Perovich et al. 2002 is relevant here since this paper does not discuss land or terrestrial snow albedos but multiyear sea ice. - p. 4, l. 3: These two references only touch on the radiation budget obliquely; citation of a paper with actual analysis such as Pistone et al. 2014 or Perovich et al., 2007 that provide actual attribution would be more appropriate. - p. 6, l. 7: change to "Figure 2. However," - p. 6, l. 12ff: The way this metric is described here and in the figure caption is confusing. Are you plotting the time period during which albedo falls into the interval {0.25,0.4}? Or are you plotting the days for which the albedo first drops below 0.4 and 0.25? Please clarify. - p. 11, l. 25 & 26: these papers are out of alphabetical order - Fig. 5: Units on vertical axis should be MJ/day

References cited: Perovich, D. K., B. Light, H. Eicken, K. F. Jones, K. Runciman, and S. V. Nghiem (2007), Increasing solar heating of the Arctic Ocean and adjacent seas, 1979 – 2005: Attribution and role in the ice-albedo feedback, Geophys. Res.

[Figure]

Lett., 34, L19505, doi:10.1029/2007GL031480. Perovich, D. K., and C. Polashenski (2012), Albedo evolution of seasonal Arctic sea ice, Geophys. Res. Lett., 39, L08501, doi:10.1029/2012GL051432. Pistone, K., Eisenman, I., and Ramanathan, V. (2014). Observational determination of albedo decrease caused by vanishing Arctic sea ice. Proceedings of the National Academy of Sciences of the United States of America, 111(9), 3322–3326. https://doi.org/10.1073/pnas.1318201111. Polashenski, C., D. Perovich, and Z. Courville (2012), The mechanisms of sea ice melt pond formation and evolution, J. Geophys. Res., 117, C01001, doi:10.1029/2011JC007231. Rösel, A., and L. Kaleschke (2012a), Exceptional melt pond occurrence in the years 2007 and 2011 on the Arctic sea ice revealed from MODIS satellite data, J. Geophys. Res., 117, C05018, doi:10.1029/2011JC007869. Rösel, A., Kaleschke, L., and Birnbaum, G.(2012b), Melt ponds on Arctic sea ice determined from MODIS satellite data using an artificial neural network, The Cryosphere, 6, 431-446, https://doi.org/10.5194/tc-6-431-2012.
* * *

---

## Author Comment (AC1) · 11 Sep 2018

Please find the reply in the attached .pdf file.

Please also note the supplement to this comment:
https://www.the-cryosphere-discuss.net/tc-2018-115/tc-2018-115-AC1-supplement.pdf
* * *

---

## Author Comment (AC2) · 11 Sep 2018

**Responses to the reviews, including relevant changes made in the manuscript, and a marked-up manuscript version**

The authors would like to thank all referees for the review of the manuscript, helpful comments, and the discussion involved in this process. The corresponding changes and improvements have been made in the revised paper and are also summarized in our reply below. Authors' responses are in blue. Reviewer's responses are in black.

**Response to Reviewer 1**

This study explores and compares the contributions of recent (satellite-era) changes in Arctic sea ice and snow cover to changes in absorbed solar energy at the surface, and finds that sea ice losses have contributed to a greater increase in solar heating than reduced Arctic snow cover. Similar analyses have been performed before, though this study carves out a niche by focusing exclusively on the Arctic (60-90 degrees) and comparing the terrestrial and sea ice contributions within this domain. An important point highlighted by the authors is that the major seasonal transition in albedo associated with sea ice loss is occurring near the summer solstice, whereas the terrestrial snow transition is shifting away from the solstice, implying greater solar forcing via sea ice loss in the recent past and near future. Overall, the paper is concise and very well-written, though there are several important aspects of the analysis that need to be revisited and/or clarified before publication. After these issues (described below) are addressed, I would support publication of this manuscript in The Cryosphere.

We thank the reviewer for their overall positive evaluation of the work and constructive comments, which were helpful in revising the manuscript. The point-by-point response to the comments is listed below.

(1) The discussion on p.7 lines 4-14 describes how lower-latitude changes in albedo drive larger changes in absorbed solar energy than equivalent albedo changes at higher latitudes. This is true for annual-mean albedo changes, but it is not true for the summer-solstice-season changes that are the focus of this section. This can be seen clearly in the authors' own Figure 5, which shows greater daily-mean solstice insolation at 80N than at 65N. The discussion on p.7 lines 4-14 is therefore largely inconsistent with Figure 5 (the latter of which is correct, I believe). The authors need to amend the discussion and re-consider potential causes of the statistics described on p.7. Differences in cloudiness and cloud trends may be a logical starting point for resolving this discrepancy

We acknowledge that the text described in your comment may have been misleading, and we have made substantial additions and alterations to the manuscript (see manuscript page 9, lines 18-26). The numbers in the paragraph are based on the instantaneous magnitude of insolation at 14:00 local solar time. We recognize, to your point, that the magnitude of the flux accumulated over the entire day around the summer solstice is larger at higher latitudes. Figure 5 was providing a simple illustration of changes in top-of-atmosphere accumulated incoming shortwave at 65° and 75°N, not the instantaneous

surface absorption. Additional text has been added explaining this figure. In the text, we calculated accumulated flux increases over ocean and land (at both latitudes) for July 1 based on the changes in albedo between 1982 and 2015 and relate the accumulated flux increases to one another.

(2) The study focuses largely on changes in net shortwave flux at the surface, and the authors acknowledge at different points in the text that changes in clouds and sea ice thickness could contribute to net shortwave changes, in addition to the more obvious contributions of changing snow and sea ice coverage. Section 2 could be improved with a bit more quantification of how large these other contributions are. Perhaps such quantification is beyond the scope of the study, though I encourage the authors to consider ways in which they could quantify the contributions of these different drivers of shortwave flux trends. It seems that the cloud contribution could be isolated and quantified via existing APP-x data, though isolating the influence of sea ice thickness/age would be more challenging.

We believe that we have improved Section 2 based on your comment. We were able to quantify the contribution of clouds on the shortwave flux at the surface by performing a calculation that determined the maximum possible effect of clouds on downwelling shortwave radiation between 1982-2015 (see manuscript page 6, lines 17-34). This calculation used the 34-year average downwelling shortwave flux for the sunlit months (March-September) and the 34-year average cloud trend (during March-September) at each point to determine the changes instantaneous shortwave flux. Further details regarding the calculation and explanation of the assumptions made may be found in the added paragraph in the text. Ultimately, we find that changes in cloud cover from 1982-2015 (during March-September) resulted in an increase of surface absorption by 1.94Wm$^{-2}$ over land and 2.19Wm$^{-2}$ over ocean. The changes account for only a 0.5% increase in incoming shortwave over ocean and 0.4% increase over land.
We regret that changes in sea ice thickness' effect on shortwave absorption cannot be explicitly determined within the scope of this study, but we hope that the theoretical maximum contribution to surface absorption changes by cloud cover has shown that changes in surface type is the dominant term in the Arctic surface energy budget.

(2b) Related to the above point, the discussion on the top of p.4 highlights a trend of lower albedo over Greenland's near-coastal regions. Is this trend caused more by reduced snow cover over the (quite small) non-glaciated portion of Greenland, or more by darkening of the perennial ice surface?

According to the average monthly trends in shortwave absorption between 1982-2015, we believe that Greenland may be decreasing in albedo due to both of the reasons you mention. Since not all of the summer months are shown in Figure 2, we have included the trends in absorption for July and August in this response (below). The spatial pattern of increased absorbed shortwave radiation over Greenland shows that the largest increases due occur in the coastal areas, but lesser increases in absorption still occur spread out over the perennially snow-covered interior- particularly in southern Greenland.

[Figure]

**July, Land Pixels    August, Land Pixels**

**Solar Radiation Absorbed (Wm$^{-2}$/year)**

-1.5   -1.0   -0.5   0.0   0.5   1.0   1.5

**Reviewer 1, Comment 2b Figure:** Trends in absorbed solar radiation at the surface from 1982-2015 for the months of July (left) and August (right) over land.

(3) A companion analysis of MERRA2 data is referred to very briefly at the end of section 2. This is a nice addition to the study, but it would be helpful if this analysis was developed more. In particular, it would be helpful to state the shortwave trends obtained from MERRA2 and show companion figures to Figs 1 and 2, so that similarities and differences between the two products can be seen more clearly. Presenting results from two or more products will give the study more credence.

We agree that the comparison with MERRA2 results was more useful when expanded. A figure showing trends from MERRA2 for June has been added as well as relevant discussion pertaining to the trends displayed in the figure (see manuscript page 7, lines 15-24). We find that the spatial agreement between the two products' output is very convincing, and thank you for your comment.

(4) Figure 5 shows that the "Low Albedo midpoint" over oceans actually occurred *before* the solstice in 2015. If this is now the norm, it implies that future trends towards earlier melt will, as with terrestrial snow, cause the high-low albedo transition to move away from the solstice. In other words, the lag between snow and sea ice melt, in combination with melt trends, may have caused the snow/sea ice

forcing differences to have *already* peaked. In light of this (and if I have interpreted correctly), the authors may want to add a bit of nuance to the abstract and associated discussion.

We believe that you have correctly interpreted that the maximum difference between the absorption changes over land and ocean has recently occurred. We agree that as the albedo transition over ocean moves further from the summer solstice, the trends over ocean will approach those over land while the difference between the trends shrinks. We added text to mention this explicitly (see manuscript page 8, lines 30-34 and page 9, lines 1-2), and have revised language elsewhere that reflects this insight.

(5) The analysis of feedbacks in section 4 should either be removed, or methodological details of this analysis need to be clarified. Personally, I think this section could simply be removed, as it does not add much to the study, and is likely sensitive to methodological choices. If it is retained, more detail is needed on methodology, including how the spatial and temporal averaging of temperature (in particular) was conducted. Were monthly or annual trends in T used? Were gridcell-level or Arctic (or global) T trends used? (e.g., how exactly were the presented June feedback numbers calculated?) Personally, I think feedback analyses are only meaningful when large areas (e.g., hemispheric or global) are used for temperature averaging. More detail would also be needed on the technique used to determine: (d alpha_p / d alpha_s).

Methodological detail has been added (see manuscript page 11, lines 3-11) and clarification of monthly, gridcell-level trends were used for analysis.

(6) Conclusions, p8,13-17: This passage could be important, but needs clarification. To be clear, Flanner et al (2011) assumed constant seasonal cycles of the albedo of multiyear and first-year sea ice. Thus from interannual changes in sea ice extent and transitions multi-year to first-year ice did contribute to area-averaged albedo changes in that study, but changes in ice albedo due to thinning or earlier ponding (etc) did not. Line 15 states that "...here we find that the inclusion of inter-annual changes to surface albedo result in a significant change to the surface shortwave energy budget...". Do the "inter-annual changes to surface albedo" refer to changes in the albedo of the ice itself? If so, I do not recall seeing quantification of this contribution in the analysis (though I think it would be very useful!). The subsequent sentence also needs clarification: "Since 2010, for example, average ocean albedo in the study area during late June has been as low as mid-September albedo in 1982-1985." Are you referring to the ocean-wide albedo, or to the albedo of the sea ice itself? If the former, this is likely due simply to the reduction in June sea ice extent, and this would have been accounted for in the Flanner et al study. If not, this is a useful finding, but one that should be reported and further developed earlier in the study.

To clarify, our changes in shortwave absorption include real (observed) albedo changes in the ocean and sea ice using the retrieved instantaneous albedo of every pixel of sea ice as well as ocean. Therefore, the mentioned inter-annual changes to surface albedo referred to both changes in the morphology of the sea ice (thinning, ponding, etc.) and also the ocean-wide albedo. This is further shown in the figure added to this manuscript that shows the differences between MERRA2 and APP-x shortwave absorption trends during June. The assumption of a constant sea ice albedo by MERRA2 versus the observed month-bymonth changes in ice albedo from APP-x illustrates the effect of including ice thickness/age and (presumably) liquid water or inhomogeneity at the surface of sea ice. It may very well be that the reduction of sea ice extent in June is a primary driver of this change, but the stronger, more positive trends in shortwave absorption in our results compared to MERRA2 indicate that there is some additional effect by the thinning or darkening of sea ice.

General: I appreciate the focus of this study on the "Arctic", defined here as 60-90N, but it is important to note that any conclusion about the relative magnitudes of sea ice vs. snow changes will be sensitive to the latitude bands selected. For example, if the latitude threshold for the analysis was adjusted equatorward, the relative contribution of land snow changes would clearly increase. I think this point should be acknowledged more clearly.

The selection of 60-90°N was intentional in hoping that we may capture the evolution of the Arctic surface over land and ocean over the satellite record. We agree, however, that analyzing the net changes in shortwave absorption from the pole to the equator could weaken or even refute our hypothesis. We attempt to make it clear in the manuscript that we are interested in the effects of changing snow and ice on the local surface energy budget, and only wish to consider pixels where the main drivers of change in the surface energy budget are such.

p1,21: A slightly clearer way to say this would be "September sea ice extent decreased by 45% from..."
p2,30: 1972 should be 1979, in reference to the Flanner et al study.
p6,22: This either needs "although" after the comma, or it should be two sentences.
p6,24-30: Please list the solstice DOY, for reference. Figure 1: It looks like there is only one point per year in this figure, so I assume you mean "Annual Absorption" instead of "Monthly Absorption".

Revisions based on the above minor comments have been made, and we thank the reviewer for their careful reading of the paper.

Figure 2: Could you speculate on the cause of the negative trend in April non-land albedo?

The negative trend in non-land albedo during April is possibly based on an increase in April cloud cover in the Arctic that spanned from the Canadian Archipelago through Alaska and into the East Siberian Sea. The increase in clouds reduced shortwave surface absorption. This increase in cloud cover did not occur over the central Arctic Ocean, however, and we cannot speculate what drove the decrease in absorbed radiation north of 80° in the Canadian-Russian Arctic.

**Response to Reviewer 2**

(1) It is stated that the time of the high-to-low albedo transition each year is moving toward the high sun of the summer solstice over ocean but moving away from the summer solstice over land. Some explanation is needed why this happens.

The manuscript was lacking a direct explanation of why snow on land (typically) melts earlier in the year than sea ice, which has been added. It has to do with higher temperatures and solar zenith angles at lower latitudes where more of the snow is compared to sea ice. Our explanation also mentions the tendency for snow-free land adjacent to snow-covered land to heat up faster than unfrozen ocean around sea ice (see manuscript page 8, lines 9-12).

(2) They claim that decreasing sea ice cover, not changes in terrestrial snow cover may play an even larger role in future Arctic climate change. The paper is not about prediction of future state of the arctic so there is no substantiation of what might happen in the future.

We have made adjustments to the abstract which removes any language referring to the changes in future ice and snow. We based this original statement on the movement of the ocean low albedo period toward the summer solstice, but we have since realized that the peak forcing of the sea ice albedo feedback may have already passed that point. The low albedo period has begun moving away from the solstice over ocean as well as land in recent years, though we expect the ice albedo feedback to remain relatively strong in the near future. The manuscript conclusion has been modified based on this explanation.

(3) There is confusion when dealing with the surface and top of the atmosphere (TOA). At TOA not only surface propertied matter but also clouds so it is a mixed signal (example to be given in Specific Comments).

The changes in shortwave absorption that were determined in the manuscript include real (observed) albedo changes at the surface using the retrieved instantaneous albedo of every pixel from the APP-x dataset. Some of this change in albedo may be due to changes in clouds, which, to your comment, makes the TOA incoming flux much different than the surface flux. A significant addition has been made to the manuscript to address the potential of cloud cover changes on absorbed shortwave at the surface (see manuscript page 6, lines 17-34). As mentioned in your specific comments, we use TOA (planetary) albedo change from the 2014 Pistone et al. study to highlight that the Arctic albedo has been generally decreasing. All other usage of the word "albedo" in the manuscript refers to albedo at the surface, unless stated otherwise.

(4) There is some lack of clarity about the impact of changes at the surface and the latitudinal changes in the solar radiation reaching the ground on the amount of absorption at the surface. How do you separate these two factors?

We do not try and normalize the differences in the amount of insolation at high vs. low latitudes in any way. Our goal is to assess the amount of absorbed solar radiation without adjusting it for the amount of incoming radiation. We hope that the text makes clear that the main driver of absorption changes is not based on latitudinal changes in insolation, but rather changes over time in surface albedo, cloud, and other variables. We interpret the "impact of changes at the surface" in your fourth comment to be concerning the changes in cloud cover or vegetation. Chapin et al. (2005) determine that on cloud-free summer days, broadband albedo has been reduced by 0.0002 per year due to changes in vegetation over the Alaskan North Slope region. Sturm et al. (2011) state that during the spring, land with more exposed shrub experienced albedo decreases earlier in the year than where there was less shrub or no vegetation at all. Some of the increasing absorption trends in the early spring over land may be caused by such changes in vegetation. Sturm et al. also point out that once temperatures became above-freezing, sensible heat flux overtook solar heating and the impact from lower albedo values was reduced, meaning that changes in absorption over land during the summer months are primarily driven by changes in snow cover, not vegetation. We have expanded the discussion on changes in vegetation and their impact on absorption (see manuscript page 5, lines 5-12) and removed redundant sentences at the end of the paragraph.

(5) MERRA-2 is also used in the analysis. Nothing is said about the differences in spatial and temporal scales of MERRA-2 and AVHRR. What impact does it have on the conclusions?

We have added a figure showing the results of MERRA2 absorption trends compared to results from APP-x and expanded the discussion of these results. Overall, the MERRA2 results are very similar to the APP-x results both spatially and with respect to the magnitude of the forcing.

Specific Comments:
Stated: Between 1979 and 2011, the Arctic top-of-atmosphere (planetary) albedo decreased from 0.52 to 0.48 (Pistone et al., 2014), and subsequent years with record or near-record low sea ice extent have further increased the amount of heat absorbed in the Arctic (Pistone et al., 2014). Comment The connection here between TOA and surface is confusing.

We have added a significant amount of explanation on manuscript pages 7-8 which talks about the difference between TOA incoming shortwave flux and accumulated flux at the surface. We also added text to the manuscript at the bottom of page five that addresses the changes in clouds and how they may cause TOA and surface downwelling shortwave flux to differ.

Stated: Snow extent has decreased over Eurasia and North America since the late 1980s Robinson & snow cover and the radiative balance over mid- and high-latitude land in the Northern Hemisphere (Groisman et al., 1994), in which retreating snow cover has led to a lower polar Comment The topic is the Arctic, so need to be focused.

Noted, but here we introduce the reader to literature that explains the trend of increasing absorption over land during spring, which our results agree with.

Stated: preconditioning of sea ice in the winter can influence the albedo into the fall of the following year, illustrating how changes in cloud cover during different seasons may affect the planetary albedo (Letterly et al., 2016; Liu & Key, 2014). Comment Above is a mixed bag of statements. Needs to be cleared.

This statement has been changed in the text. We intended to say that besides the effects of clouds on instantaneous absorption at the surface, they can influence sea ice changes, which in turn affects the non-land albedo. Thank you for bringing it to our attention!

Stated: This study focuses on the effects of snow and ice cover changes on the surface shortwave radiation budget of the Arctic - defined as the area poleward of $60 \circ N$ - not the remote effects of mid-latitudes on the Arctic Comment Why to invoke remote effects of mid-latitudes on the Arctic? This topic is not of relevance here and the comment does not add much to the discussion.

Other works, such as that of Flanner et al. (2011), perform a similar experiment but include the effects of changes in snow cover at lower latitudes. These changes in mid-latitudes are not considered in this study, which we mention here.

Stated: Furthermore, since terrestrial snow cover has mostly melted by June, the main drivers of absorption trends over land during the summer may be changes in cloud cover or vegetation (Chapin et al., 2005; Loranty et al., 2011). Comment: There is no discussion of changes in vegetation after melt and its impact on absorption.

Some discussion of these factors was in the original manuscript. Additional text on the effects of changing vegetation and its impact on broadband albedo has been added.

Stated: In contrast, most of the sea ice lasts through early summer, but changes in sea ice thickness still allow for changes in absorption (Perovich & Polashenski, 2012). Comment: The impact of ice thickness on absorption was not addressed in this paper. Needs more discussion.

We agree that the impacts of sea ice thickness on shortwave absorption are very important, and believe that they are a large reason for our absorption trend results from APP-x (which includes the effects of ice thickness in the surface albedo). Text and an additional reference addressing the changes of albedo with sea ice thickness has been added (see manuscript page 5, lines 18-19).

**Response to Reviewer 3**

(1) The analysis and discussion of albedo differences and trends over sea ice (beginning on p. 4, bottom) is much too simplistic and misleading, since it implies that seasonal evolution and inter annual trends are driven entirely by changes in ice thickness. However, prior to the onset of melt the differences in snow albedo on different ice classes (with the exception of very thin ice) are likely insignificant. Much more important in this context are development, areal fraction, and optical properties of ponds on sea ice. It is here that major contrasts between different ice age (MY vs. FY ice) and ice thickness are expressed. Moreover, these processes are dominant for the months of June and July, such that the discussion of Fig. 2 (June) in terms of ice thickness classes is not really appropriate. Here, a more rigorous discussion of the observed trends in terms of the seasonal cycle of ponds on sea ice (Perovich and Polashenski, 2012; Polashenski et al., 2012; Rösel and Kaleschke, 2012a) is needed. In particular, the work by Rösel and Kaleschke (2012a,b) is highly relevant because it discusses the role of spatio-temporal variations in ponding on sea ice in the context of sea ice concentration and extent anomalies based on remote sensing data. A closer examination of their findings may help explain some of the spatial patterns seen in Fig. 2 and the inter annual variations shown in Fig. 1. It is also relevant in the discussion of reductions in albedo in the month of June (p. 4, l. 18) which is as much or more a function of ponding as of reduction in ice concentration.

Thank you for the insightful comment. While we believe that changes in mean Arctic ice thickness have some effect on annual trends, we have added a number of sentences to the analysis and discussion of absorption trends which include the process of melt pond formation and its effect on sea ice albedo (see manuscript page 5, lines 16-24). In the revised manuscript, we attribute the changes in absorption over ocean (especially in summer) to changes in sea ice area/extent as well as to the occurrence of melt ponds on the increased amount of first-year ice. Both Rösel references have been added.

(2) The authors attribute changes in albedo over land to changes in snow cover duration during the snow-covered period, and to changes in vegetation and cloudiness after loss of snow cover. This may be too simplistic and requires further analysis. First, while the albedo contrasts between clouds and snow cover may not be as large as those between land surface and clouds, they cannot be ruled out as important without further analysis. For example, the spatially coherent trend for the month of April towards reduction in absorption of shortwave energy in NW Siberia may well be due to changes in cloudiness rather than duration of snow cover (which persists well beyond April). Without specific references to the published literature or some additional analysis of a particular subregion in a case study it is difficult to accept the explanation offered by the authors wholesale.

We have added a significant analysis describing the potential effects of cloud cover on surface absorption over the study period (see manuscript page 6, lines 17-34). We examined the trend of increased absorption over NW Siberia and found that cloud cover decreased ~1% from 1982-2015 in that region which corresponds to an increase in shortwave absorption at the surface. The figures inserted below show the trends described. While the changes in cloudiness over Northwestern Siberia may have impacted the overall trend, we see that the decrease in surface absorption due to clouds was outweighed

by changes in surface absorption due to other factors, possibly increased snow cover. We hope that the analysis of potential cloud effects added to the manuscript offers enough explanation that the reader can accept that changes in the snow/ice surface is the main driver of these changes.

[Figure]

**Reviewer 1, Comment 2 Figure:** Changes in April cloud trends (left) and changes in April surface shortwave absorption (right) 1982-2015. Data shows a decrease in cloud cover over NW Siberia which corresponds to an increase in absorbed shortwave radiation.

(2b) Second, some of the discussion of snow and ice albedo variations needs to be reviewed and potentially revised. For example, on p. 4, l, 11ff a difference between snow albedo over sea ice (0.6) and land (0.7) is seen as being important. Where do these estimates come from? The paper by Sturm et al. cited here only discusses tundra snow. Both values are low if they refer to dry early spring snow before the onset of melt. Moreover, I am not aware of data showing that the albedo of snow covered sea ice is that much lower than that of tundra snow. This needs to be either corrected or further substantiated.

These estimates originally came from the cited Sturm et al. paper. We have updated the values to 0.8 dry, snow-covered sea ice (Rösel et al., 2010) and 0.85 (Greenfell and Perovich, 2004). These references have been included. The key is the difference between open land and open water. The albedo difference between open water and sea ice is and will continue to be more impactful than the albedo difference between snow-covered land and bare land.

(3) The discussion of the spatial patterns of trends could be expanded a bit, ideally by referencing either published work or at least a slightly more in depth analysis for a subregion.

We have added detail to this section. Particularly, we highlight the changes in absorption over the marginal Arctic Seas compared to those over high-latitude land masses (see manuscript page 5, lines

27-28). We note that our absorption trend results match the spatial patterns and (relative) magnitudes of changes in radiative forcing over land and ocean seen by Flanner et al. (2011).

(3) It is asserted that spatially coherent trends such as that in Greenland or Siberia are driven by trends in cloudiness. This appears plausible, but would benefit from some more detail. However, this raises the question as to what spatially heterogeneous trends are driven by (e.g., April over much of the landmass, or June in much of North America). Are the trends for individual grid cells or small aggregations of grid cells significant if they are neighboring on grid cells with opposite sign in the trend?

We have added text expanding the effects of cloud trends. In the above responses to comments, we include an image that shows the trend in April cloud cover from APP-x data over land. The spatial patterns in the surface absorption trends, then, are likely driven by the changes in cloud trends, which fit the description you give above. While not as obvious as heterogeneity depicted in April and June over North America, we point to Figure 3 in the manuscript, which shows that many cells with adjacent, oppositely-signed absorption trends during September correspond well with changes in clouds (i.e., south of the Laptev Sea in coastal and interior Siberia).

(4) The previous comment relates to a significant shortcoming in the manuscript that should be easily remedied. Specifically, the discussion of the methods employed in deriving the different data sets and their analysis is currently much too superficial. First, it would be preferable to separate the description of the datasets used and the analysis methods employed from the reporting of results. Specifically, l. 24 on p. 3 would be a natural break. Then, while reference to the paper by Key et al. (2016) to describe the data product is fine, the current paper needs to provide more information on how the data sets were generated in particular as relevant to the specific variables (albedo and absorbed shortwave energy, for example) discussed here. For example, how has broadband albedo been derived from spectral radiances and what are the associated errors and uncertainties associated with such a derivation? With the ocean regions located at higher latitudes than the land areas, does this introduce a potential bias because of lower solar zenith angles relative to sensor zenith angles? With a lack of bireflectance distribution function (BRDF) data over melting sea ice as opposed to snow cover over land this may introduce significant uncertainties as well. Further details for the MERRA reanalysis products should also be provided in a restructured methods and data section.

We have redesigned and improved the section explaining the APP-x dataset (manuscript page 3, lines 20-31) and have separated the explanation of the data from the analysis of results. There is now a paragraph that details how albedo and downwelling shortwave radiation were determined within the APP-x dataset as well as their uncertainties and limitations. To answer your specific questions: Key et al. (2001) states that while the anisotropic reflectance correction is not perfect, it does account for potential viewing and illumination biases. More detail on the MERRA2 analysis, including results and input variables, has been added.

Specific edits: - p. 1, l. 4: For clarity throughout it may be better to refer to sea ice albedo feedback and snow albedo feedback (or at least clarify at the start that ice albedo feedback only refers to sea ice)

Done.

- p. 1, l. 12: reads a bit awkward, maybe change to something like "the timing of the seasonal transition from high to low albedo . . . shifting towards greater insolation associated with summer solstice"

Re-worded, thank you for the suggestion.

 - p. 3, l. 27: change to "confidence of the trends is calculated"

Changed.

 - p. 4, l. 2: Not sure how Perovich et al. 2002 is relevant here since this paper does not discuss land or terrestrial snow albedos but multiyear sea ice.

This reference was removed.

- p. 4, l. 3: These two references only touch on the radiation budget obliquely; citation of a paper with actual analysis such as Pistone et al. 2014 or Perovich et al., 2007 that provide actual attribution would be more appropriate.

The Serreze et al. reference was removed and the Pistone et al. reference has been added.

 - p. 6, l. 7: change to "Figure 2. However,"

Edited.

- p. 6, l. 12ff: The way this metric is described here and in the figure caption is confusing. Are you plotting the time period during which albedo falls into the interval {0.25,0.4}? Or are you plotting the days for which the albedo first drops below 0.4 and 0.25? Please clarify.

The latter. We are plotting the days for which the albedo first drops below 0.4 and then below 0.25. This has been clarified in the text.

 - p. 11, l. 25 & 26: these papers are out of alphabetical order

Fixed.

 - Fig. 5: Units on vertical axis should be MJ/day

Changed. Thank you for the insight and close reading, with all of your comments.

[revised manuscript text omitted]